# Photodynamic Inactivation of Foodborne Bacteria: Screening of 32 Potential Photosensitizers

**DOI:** 10.3390/foods13030453

**Published:** 2024-01-31

**Authors:** Amritha Prasad, Erin Wynands, Steven M. Roche, Cristina Romo-Bernal, Nicholas Allan, Merle Olson, Sheeny Levengood, Roger Andersen, Nicolas Loebel, Caetano P. Sabino, Joseph A. Ross

**Affiliations:** 1Chinook Contract Research Inc., Airdrie, AB T4A 0C3, Canada; amritha@ccr01.com (A.P.); nick.allan@ccr01.com (N.A.); merle.olson@avetlabs.com (M.O.); 2ACER Consulting, Guelph, ON N1G 5L3, Canada; ewynands@acerconsult.ca (E.W.); sroche@acerconsult.ca (S.M.R.); 3Ondine Biomedical Inc., Bothell, WA 98011, USA; cromo@ondinebio.com (C.R.-B.); slevengood@ondinebio.com (S.L.); randersen@ondinebio.com (R.A.); nloebel@ondinebio.com (N.L.); csabino@ondinebio.com (C.P.S.); 4Center for Lasers and Applications, Energy and Nuclear Research Institute, São Paulo 05508-000, SP, Brazil

**Keywords:** foodborne pathogens, biofilms, food safety, photodynamic disinfection, antimicrobial photodynamic therapy, food decontamination

## Abstract

The development of novel antimicrobial technologies for the food industry represents an important strategy to improve food safety. Antimicrobial photodynamic disinfection (aPDD) is a method that can inactivate microbes without the use of harsh chemicals. aPDD involves the administration of a non-toxic, light-sensitive substance, known as a photosensitizer, followed by exposure to visible light at a specific wavelength. The objective of this study was to screen the antimicrobial photodynamic efficacy of 32 food-safe pigments tested as candidate photosensitizers (PSs) against pathogenic and food-spoilage bacterial suspensions as well as biofilms grown on relevant food contact surfaces. This screening evaluated the minimum bactericidal concentration (MBC), minimum biofilm eradication concentration (MBEC), and colony forming unit (CFU) reduction against *Salmonella enterica*, methicillin-resistant *Staphylococcus aureus* (MRSA), *Pseudomonas fragi*, and *Brochothrix thermosphacta*. Based on multiple characteristics, including solubility and the ability to reduce the biofilms by at least 3 log_10_ CFU/sample, 4 out of the 32 PSs were selected for further optimization against *S. enterica* and MRSA, including sunset yellow, curcumin, riboflavin-5′-phosphate (R-5-P), and erythrosin B. Optimized factors included the PS concentration, irradiance, and time of light exposure. Finally, 0.1% *w*/*v* R-5-P, irradiated with a 445 nm LED at 55.5 J/cm^2^, yielded a “max kill” (upwards of 3 to 7 log_10_ CFU/sample) against *S. enterica* and MRSA biofilms grown on metallic food contact surfaces, proving its potential for industrial applications. Overall, the aPDD method shows substantial promise as an alternative to existing disinfection technologies used in the food processing industry.

## 1. Introduction

Effective food safety practices play a pivotal role in safeguarding public health. The Centers for Disease Control and Prevention estimate that foodborne pathogens cause 48 million cases of illness and 3000 deaths in the United States annually [1]. Similarly, the Public Health Agency of Canada estimates that there are 4 million episodes of foodborne illness in Canada annually [2]. The pathogens that most commonly cause illness are, in descending order of incidence, Norovirus, nontyphoidal *Salmonella* spp., *Clostridium perfringens*, *Campylobacter* spp., and *Staphylococcus aureus*, while the pathogens that most frequently cause hospitalization are nontyphoidal *Salmonella* spp., Norovirus, *Campylobacter* spp., *Toxoplasma gondii*, and Shiga toxin-producing *Escherichia coli* [3,4].

From a system-wide perspective, microbial control strategies for food production are highly complex. Along the production chain, microbial populations found in water sources, animal microbiota, and food processing and packing workflows can potentially lead to extensive pathogen dissemination in the globalized economy. The One Health perspective emerges in this context as a vital paradigm, acknowledging the intricate interplay between human health, animal health, and environmental integrity in safeguarding our food supply [5]. Furthermore, this initiative addresses the concerning emergence of antimicrobial resistance caused by excessive antibiotic usage in both human healthcare and animal production, with potentially catastrophic consequences to future approaches to infection control [6]. Therefore, a critically important goal of pathogen reduction within food production and processing systems is to avoid the promotion of antimicrobial resistance to further safeguard human health [7]. In addition to causing widespread illness, microbial food contamination also leads to food spoilage, necessitating food recalls, significant economic losses, and further food safety concerns [8]. *Pseudomonas fragi* and *Brochothrix thermosphacta* are the most prevalent food spoilage organisms in foods packaged aerobically, vacuum sealed, or with a modified atmosphere [9,10,11]. Hence, efforts to reduce microbial contamination aim to not only protect human health but also to extend the food shelf life and maintain nutritional and organo-sensorial quality [12,13].

Biofilms pose a significant challenge in the food industry and are of particular concern in meat packing plants where the disinfection of food contact surfaces is essential [14]. In brief, biofilms consist of surface-attached, complex microbial ecosystems comprising one or more species embedded within a polymeric extracellular matrix [15]. These robust structures are often impermeable to chemical biocides and create a source of pathogens that are intrinsically resistant to conventional disinfection methods [15]. Additionally, the presence of multiple clones and species within biofilm communities facilitates the vertical and horizontal transfer of antimicrobial resistance- and virulence-related genes that potentiates the formation of new clonal complexes of multidrug-resistant strains [16]. Biofilms grown on surfaces within food processing facilities can shed planktonic organisms that can contaminate and colonize other downstream surfaces, such as industrial equipment and utensils, and the food product itself [17,18]. Therefore, when novel multidrug-resistant clones emerge from biofilms found in food production facilities, a rapid environmental dissemination could eventually result in community and nosocomial outbreaks throughout the affected population [6].

Standard decontamination practices, such as using sanitizing agents or heat treatments, are routinely implemented in food processing to inactivate or mechanically remove microorganisms from food contact surfaces and food matrices [19]. Pathogen control in meat processing facilities requires a comprehensive approach, ranging from minimizing contamination on live animals to minimizing the transfer of microorganisms to carcasses, reducing microbial levels on meat, and inhibiting the growth of microorganisms [20]. However, existing biocidal chemicals and technologies can lead to the development of microbial resistance and leave residues affecting the quality of food products [21,22]. Some microorganisms develop tolerance responses to stress conditions imparted by the antimicrobial practices by means of gene modifications or the development of biofilms [23,24]. Moreover, chemicals used for disinfection can have corrosive effects on hard surfaces, cause eye and skin irritation for personnel, and alter food product organoleptics [25,26]. Emerging nonthermal technologies like cold plasma technology, high intensity ultrasound, and light-based technologies have shown promise for the decontamination of food contact surfaces [19].

Antimicrobial photodynamic therapy, or photodynamic disinfection (aPDD), has a long medical history in the treatment of localized infections and decontamination of blood products [27,28]. Photodynamic disinfection provides a promising method to safely sanitize food contact surfaces without the use of harsh antimicrobial chemicals, ionizing radiation, or heat [29]. aPDD involves the administration of a non-toxic light-sensitive substance, known as a photosensitizer, followed by exposure to visible light at a specific wavelength. After absorbing photons of visible light, the photosensitizer molecule undergoes a photophysical process producing reactive oxygen species (ROS) that are lethal to microorganisms [30]. aPDD produces its antimicrobial effect by means of photodynamic inactivation (PDI) that includes two types of reactions: Type 1 and Type 2. Type 1 includes electron transfer, gated by the PS, from the neighboring molecule to surrounding oxygen, leading mainly to the formation of a superoxide radical. In Type 2, upon irradiation, the excited PS transfers the energy directly to the surrounding oxygen molecules to form singlet oxygen species [31]. The produced ROS can further produce cytotoxic reactions in the bacterial cells, such as cell membrane damage, DNA damage, and DNA oxidation [30,31]. Following the cessation of light exposure, such reactive oxygen species are rapidly consumed or quenched and leave no toxic residues on treated materials. Due to the immediate, broad-spectrum microbicidal effect, aPDD could provide meat processing plants with a disinfection tool that enhances quality of the food product without the downside of by-product residues from sanitizers or contributing to the development of antimicrobial resistance [21]. Recent studies have investigated the efficacy of aPDD in reducing microorganisms on both food contact surfaces and food products [32,33]. In addition to improving food processing methods to reduce or eliminate the pathogen load, aPDD can also be used to extend shelf life by eliminating microorganisms associated with food spoilage [34]. aPDD has also been used to treat food processing workers themselves, specifically within the anterior nares, to reduce carriage of host-borne microbes that present high transmission risk to food products, food-processing surfaces and other workers [35]. 

In the present study, we explored the efficacy of aPDD against biofilms produced by disease- and spoilage-causing microorganisms. A total of 32 food-safe pigments, including several Health Canada-approved food coloring agents, were tested as potential photosensitizers (PSs) in combination with exposure to light of respectively suitable wavelengths, and the performances of the top candidates were tested against biofilms grown on materials typically used in food contact surfaces.

## 2. Materials and Methods

### 2.1. Preparation of Candidate Photosensitizers

A total of 32 PS candidates comprised of natural products and Health Canada-approved food colorants were procured from multiple vendors (listed in Table 1). Master stocks of the PS candidates were prepared at various concentrations (Table 1; see Appendix A for a listing of tested concentrations in both % *w*/*v* and mM) via dissolution in EP (European Pharmacopoeia)/JP (Japanese Pharmacopoeia)/USP (United States Pharmacopoeia)-purified water (Ricca, Arlington, TX, USA), dimethyl sulfoxide (DMSO; Sigma life sciences, St. Louis, MO, USA), or propylene glycol (Aldrich Chemical Company Inc., Milwaukee, WI, USA). Titanium dioxide did not form a homogenous solution in water, DMSO, or in propylene glycol; therefore, a 1% (*w*/*v*) master stock of titanium dioxide was prepared by suspending powdered PS in water and sonicating for 30 min in a stainless-steel insert tray in a water bath style sonicator (VWR 550T, Avantor Sciences, Radnor, PA, USA), followed by storage at −80 °C (Thermo Scientific, Mississauga, ON, Canada). Additionally, turmeric extract did not form a homogenous solution in water or DMSO but instead was dissolved to 0.1% *w*/*v* in propylene glycol and stored at −80 °C. The master stocks were then diluted 10-fold in water (EP/JP/USP pure) and loaded on the UV-Vis 96-well plates. The PSs in the 96-well plates were further diluted by 10- and 100-fold in water (EP/JP/USP-purified). Water and 0.1% *v*/*v* DMSO in water were used as blanks. The total volume of the contents in each well was 180 µL. The absorbance of the plates was analyzed over the wavelength range of 200–800 nm using a UV-Vis spectrophotometer (Synergy HTX multi-mode reader, model S1Lf, BioTek, Winooski, VT, USA). The spectral scans of the PS candidates are shown in Appendix A. The highest wavelength yielding an obvious peak, or the wavelength yielding a predominant peak, was used for analyzing the antibiofilm and antimicrobial effect of the antimicrobial photodynamic disinfection (aPDD) condition in this study (Table 1).

### 2.2. Light Emitting Diode (LED) System

The LED system consisted of a controller (LUM2CON, LumiDox Gen2, Analytical Sales and Services Inc., Flanders, NJ, USA) and 96-well LED array sources (LumiDox Gen2, Analytical Sales and Services Inc., Flanders, NJ, USA) emitting light of wavelengths 660 nm (Deep red, LUM296LA660), 630 nm (Red, LUM296LA630), 590 nm (Amber, LUM296LA590), 527 nm (Green, LUM296LA527), 505 nm (Cyan, LUM296LA505), 470 nm (Blue, LUM296LA470), 445 nm (Indigo, LUM296LA445), 420 nm (Violet, LUM296LA420), 365 nm (Ultraviolet-A, UV-A, LUM296LA365), and warm white light (Table 1). The LED treatments were performed by placing the multi-well test devices directly onto the 96-well LED arrays. The irradiance of the light emitted from the LED source could be set in multiple power stages ranging from 1–5, each stage corresponding to a specific irradiance value for each LED source. For screening of the PSs, the LED system was used with the irradiance corresponding to stage 5 (highest) corresponding to the irradiance values reported in Table 1, and the treatment time was 8 min with each LED corresponding to a specific light dose (i.e., radiant exposure expressed in J/cm^2^) as listed in Table 1.

For further optimization of PS candidates, riboflavin-5′-phosphate (R-5-P), sunset yellow, and curcumin light treatments were performed with blue (470 nm) LEDs set at 170 mW/cm^2^ of irradiance for 1, 2.5, and 5 min, corresponding to treatment doses of 10.2, 25.5, and 51 J/cm^2^, respectively (Appendix A). The treatment for erythrosin B was performed with the LED emitting light of wavelength 527 nm (Green) for 2, 5, and 10 min corresponding to treatment doses of 10.2, 25.5, and 51 J/cm^2^, respectively, at an irradiance value of 85 mW/cm^2^ (Appendix A). The antibiofilm effect of R-5-P against biofilms formed on selected food contact surfaces was assessed for a 445 nm (Indigo) LED treatment for 5 min (55.5 J/cm^2^) at an irradiance of 185 mW/cm^2^ (Appendix A). All the LED treatments were performed in the dark to minimize interference from the surrounding light.

### 2.3. Bacterial Culture Preparation and Biofilm Growth

*Salmonella enterica subs. enterica sv.* Choleraesuis (ATCC 10708), methicillin-resistant *Staphylococcus aureus* (MRSA 456 S. Yanke), *Pseudomonas fragi* (ATCC 4973), and *Brochothrix thermosphacta* (ATCC 11509) were used in this study. Tryptic soy agar plates (TSA; PT80, Dalynn Biologicals Inc., Calgary, AB, Canada) were used to restore the frozen stocks of each microorganism by streaking their primary cultures. The streaked TSA plates were incubated at 37 °C in the case of *S. enterica* and MRSA, while they were incubated at 26 °C in the case of *P. fragi* and *B. thermosphacta*, for 24 h. This was followed by streaking a secondary culture on TSA plates from the primary streaks, which were incubated as above. Isolated colonies from the secondary cultures were used to inoculate tryptic soy broth (TSB; BT85, Dalynn Biologicals Inc., Calgary, AB, Canada), yielding saturated “overnight” cultures after 18–24 h at 37 °C (for *S. enterica* and MRSA) or 24–48 h at 26 °C (for *P. fragi* and *B. thermosphacta*) based on the growth conditions required for each organism tested.

The overnight cultures were diluted 10,000-fold in TSB to prepare diluted inocula for each of the four microbial suspensions, and 150 µL of the diluted inocula was used to inoculate the minimum biofilm eradication concentration (MBEC) devices (Innovotech Inc., Edmonton, AB, Canada). These devices were then incubated at 37 °C for *S. enterica* and MRSA (or at 26 °C for *P. fragi* and *B. thermosphacta*) with gentle shaking at 110 rpm for 24 h to facilitate biofilm growth on the polystyrene pegs of the MBEC devices. The overnight cultures and the diluted inocula were enumerated via serial dilution in 0.9% saline (Cytiva, HyClone Laboratories, West Jordan, UT, USA) and spot plating onto TSA plates to confirm inoculum densities. The initial count of colony forming units (CFUs) of the overnight cultures and diluted inocula were 10^9^ CFU/mL and 10^5^–10^6^ CFU/mL, respectively.

### 2.4. Screening of Photosensitizer Candidates against Biofilms and Planktonic Cells

The screening process for the 32 PSs outlined in Table 1 and Appendix A was conducted essentially according to the protocol described in Harrison et al. [37]. The working stocks of the PS candidates were prepared by diluting the master stocks by either 10-fold (for aqueous master stocks in water) or by 100-fold (for organic master stocks prepared in either DMSO or propylene glycol) in water. For the treatment of biofilms with the PS candidates, the 96-well plates (referred to as “challenge plates”) were loaded with 200 µL of the working stocks of the PSs in triplicate wells. Additionally, the working stocks of the PS candidates were diluted by 10- and 100-fold in water and were loaded onto the challenge plates in triplicate wells. For the PS master stocks prepared in water, 200 µL of water was used as the vehicle growth control (VC), and for the PS master stocks prepared in DMSO and propylene glycol, 100-fold diluted DMSO and propylene glycol in water were used as the VC. For sterility controls, some pegs of the MBEC devices were inoculated with 150 µL of sterile TSB and then treated with 200 µL of the working stocks or their corresponding VCs. For the screening of the PS candidates, the biofilms formed on MBEC pegs were rinsed with 200 µL of 0.9% saline for 2 min, followed by exposure to challenge plates for 5 min in the dark for the uptake of PSs by the biofilms. This was followed by treatment with the LEDs emitting light of peak wavelengths closest to the predominant absorption peak (λ_max_) of each PS candidate for 8 min corresponding to the treatment doses mentioned in Table 1. Simultaneously, an MBEC lid immersed in the challenge plate for 13 min in the dark after the rinsing step acted as a “dark control” to evaluate the dark toxicity of the PS candidates in the absence of the LED light treatments. After the challenge step, 20 µL of the contents of each well of the challenge plates was transferred to another 96-well plate (VWR International, Mississauga, ON, Canada) containing 180 µL of TSB in each well (considered the “outgrowth plate”). These plates were then incubated at 37 °C (*S. enterica*, MRSA) or 26 °C (*P. fragi*, *B. thermosphacta*) for 24 h with gentle shaking. The treated pegged lids were rinsed in 0.9% saline for 2 min and were transferred to a 96-well plate containing the recovery broth (“recovery plate”), comprised of cation-adjusted Mueller–Hinton Broth (MHB; Becton, Dickinson and Company, Sparks, MD, USA) supplemented with a neutralizer supplement, consisting of L-histidine (VWR International, Mississauga, ON, Canada), L-cysteine (VWR International, Mississauga, ON, Canada), and reduced glutathione (Sigma Aldrich, Winston Park Drive, Oakville, ON, Canada). The recovery plates were then sonicated in a water bath sonicator (Branson M8800, ITM Instruments Inc, Montreal, QC, Canada) for 30 min for the recovery of the biofilms. After sonication, 100 µL from each well was serially diluted and spot plated on TSA plates and incubated as above for the enumeration of CFUs. For the semiquantitative determination of MBEC values, 100 µL of TSB was added to each well of the recovery plates and incubated at 37 °C (*S. enterica*, MRSA) or 26 °C (*P. fragi*, *B. thermosphacta*) with gentle orbital shaking for 24 h. After incubation, a semiquantitative assessment of the outgrowth and recovery plates was performed by scoring these plates visually based on turbidity, where minimum bactericidal concentration (MBC) and MBEC values were determined as the lowest concentration of the PSs that yielded no growth in the wells of the outgrowth and recovery plates, respectively. Additionally, optical density (OD) values of the outgrowth and recovery plates were analyzed at 650 nm using an Epoch microtiter plate reader (Synergy HTX multi-mode reader, model S1Lf, BioTek, Winooski, VT, USA). Clear wells corresponding to OD_650_ values of less than 0.05 indicated inhibition after the incubation period. The TSA plates were enumerated after incubation and the log_10_ CFU reductions per peg calculated relative to the corresponding vehicle controls.

### 2.5. Optimization of Photodisinfection Conditions

To refine the PS concentration and the antimicrobial photodynamic disinfection (aPDD) conditions involving blue (470 nm) and green (527 nm) LEDs, the PS candidates sunset yellow, curcumin, riboflavin-5′-phosphate (R-5-P), and erythrosin B were investigated. These PS candidates were selected based on multiple characteristics, such as the formation of a uniform solution in their respective solvents, stability of the solution formed (i.e., no precipitation during storage), and ability to reduce the biofilms by at least 3 log_10_ CFU/peg—or, where the dark VCs yielded less than 3 log_10_ CFU/peg, a “max kill” (corresponding to no colonies obtained from the treated biofilms). Riboflavin-5′-phosphate and riboflavin are chemically almost identical, with the sole difference being a phosphate group to enhance water solubility [38,39].

The optimization was also performed based on the method described in the screen above. The concentrations of the PS candidates tested were in the range of 1 to 0.00001% *w*/*v* for the PSs R-5-P, sunset yellow, and erythrosin B and 0.001–0.00000001% *w*/*v* in the case of curcumin. Two hundred microliters of sterile water was used as the VC in the case of R-5-P and sunset yellow, while a 1:100 dilution of DMSO in water was used as the VC for curcumin.

The biofilms of *S. enterica* and MRSA were formed on the MBEC pegs as mentioned in Section 2.3 with 200 µL of diluted inocula of 10^5^–10^6^ CFU/mL followed by incubation at 37 °C for 24 h with gentle shaking. After the biofilm formation, each well of the MBEC lids was rinsed with 225 µL of 0.9% saline for 2 min, followed by the immersion of the MBEC lids in the challenge plates comprising of 225 µL of the individual PS candidates with the appropriate concentrations of R-5-P, sunset yellow, and curcumin for 5 min in the dark to facilitate the uptake of PS by the biofilms, followed by the LED treatment with 470 nm (Blue) light for 1, 2.5, and 5 min corresponding to treatment doses of 10.2, 25.5, and 51 J/cm^2^, respectively. Simultaneously, an MBEC lid was immersed in the challenge plate for 10 min in the dark (corresponding to the greatest LED treatment time after the initial dark contact time). Following the challenging step, 20 µL from each well of the challenge plate was mixed with 180 µL TSB in another 96-well plate (“outgrowth plate”) and incubated at 37 °C with gentle shaking for 24 h. After the treatment, the MBEC lids were rinsed with 225 µL 0.9% saline for 2 min, followed by immersion in 225 µL of recovery broth and sonication in a digital water bath sonicator for 30 min for the recovery of the biofilms from the MBEC pegs. One hundred microliters from each well of the recovery plate was further used for serial dilution and spot plating on the TSA plates, and the plates were incubated at 37 °C for 24 h for enumeration. The cell counts were recorded in CFU/peg. The recovery plate was topped up with 125 µL of TSB and incubated at 37 °C with gentle shaking for 24 h. Similarly, in the case of erythrosin B, the biofilms on the MBEC lids were rinsed and challenged with erythrosin B, followed by treatment with a 527 nm LED for 2, 5, and 10 min. A dark control involving incubation of the challenge plate for 15 min (corresponding to the highest treatment time) protected from light was also incorporated. The challenged MBEC lids were then rinsed and recovered as above. The incubation times for the outgrowth, recovery, and TSA plates matched the procedure outlined in Section 2.4.

For evaluating the antibiofilm effect of the pairwise combination of the selected PS candidates, 2% *w*/*v* master stocks of R-5-P and sunset yellow in water and a 0.1% *w*/*v* master stock of curcumin in DMSO were prepared. The master stock of curcumin was further diluted 50-fold in water to yield a working stock of a 0.002% *w*/*v* concentration. For preparing the pairwise combination of PSs, equal volumes of 2% *w*/*v* of R-5-P were mixed with either 2% *w*/*v* of sunset yellow or a 0.002% *w*/*v* solution of curcumin, and a 2% *w*/*v* solution of sunset yellow was mixed in equal parts with 0.002% *w*/*v* curcumin. The challenge plates comprised 225 µL of PSs in pairwise combination, while 225 µL of water was used as the VC for the R-5-P and sunset yellow combination, and a 1:100 dilution of DMSO in water was used as the VC in the case of other two PS combinations involving curcumin. The biofilms for *S. enterica* and MRSA were formed on the MBEC pegs as mentioned in Section 2.3. These MBEC devices were then rinsed, challenged, treated with 470 nm (blue) LED, rinsed, and recovered via sonication as mentioned above for individual PSs. The incubations of the outgrowth, recovery, and TSA plates were performed as mentioned in Section 2.4. All the outgrowth and recovery plates were subjected to semiquantitative assessment, while the TSA plates were enumerated after incubation, and the results were reported in CFU/peg.

### 2.6. Photodisinfection of S. enterica and MRSA Biofilms on Representative Food Contact Surfaces

For this study, riboflavin-5′-phosphate was selected due to its consumption as a nutritional supplement, palatability, and general consumer acceptance. The antibiofilm effect of 0.1% *w*/*v* of riboflavin-5′-phosphate, in combination with a 445 nm (indigo) LED, was tested against a range of surfaces, including Aluminum (Al), stainless steel (SS), galvanized steel (GS), mild steel (MS), copper, and wood. These surfaces were selected in this study, as they are commonly used surfaces found in the food manufacturing and packaging industry. For example, surfaces like Al, SS, MS, and GS are most commonly used in conveyor belts, while wood is commonly used as a material for making chopping boards in the packaging plant and at home, while copper is commonly used in containers. Biofilm Surface Test Materials (BSTM^™^) devices were crafted by attaching the autoclaved test surfaces onto the lid of a conventional 12-well plate (Cellstar, Greiner bio-one North America Inc., Monroe, NC, USA) using a hot glue gun (Figure 1). The metal surfaces (Al, SS, GS, and MS) had measurements of 11 mm (length) × 6 mm (width) × ~1 mm (thickness). Standard 0.5-inch copper end caps procured from a local hardware store had measurements of 16 mm (inner diameter), 18 mm (outer diameter), and 15 mm (height). The positive growth control utilized round poplar dowel wood pieces with measurements of 0.25-inch in diameter and 11 mm in length.

For the biofilm growth, the overnight cultures of *S. enterica* and MRSA were first prepared as mentioned in Section 2.3, yielding cell counts of ~10^9^ CFU/mL, and were diluted by 10,000-fold in biofilm growth media (BGM) comprising 20% *v*/*v* TSB, 20% *v*/*v* serum (Fetal Bovine serum, HyClone GE, South Logan, UT, USA), and 60% *v*/*v* phosphate-buffered saline (pH 7.4, Cytiva, HyClone Laboratories Inc., West Jordan, UT, USA) to yield a cell count of 10^5^–10^6^ CFU/mL. Before inoculation, the test devices were rinsed with sterile water (4.2 mL for most surfaces, 2.5 mL for copper) for 2 min, followed by pre-soiling the devices with 20% *v*/*v* serum (Bovine Calf Serum, HyClone GE, South Logon, UT, USA) for 90 min. Subsequently, surfaces were inoculated with diluted inoculum (4.2 mL for most surfaces, 3 mL for copper) and incubated at 37 °C with gentle shaking for 24 h. The sterility controls were inoculated with BGM and subjected to the same incubation conditions. The challenge plates included 0.1% *w*/*v* of riboflavin-5′-phosphate as a test agent (4.5 mL per well for most surfaces, 3 mL for copper) and an equal volume of water as a vehicle growth control. After biofilm formation, the devices were rinsed with 4.5 mL of (or 3 mL) 0.9% saline for 2 min, followed by incubation in challenge plates for 5 min in the dark and treatment with a 445 nm (indigo) LED for 5 min (55.5 J/cm^2^). The test devices challenged with vehicle growth controls were incubated for 10 min in the dark. After the treatments, the test devices were again rinsed with 0.9% saline (4.5 or 3 mL) for 2 min, and the biofilms were recovered in recovery broth via sonication in the digital water bath sonicator for 30 min. After sonication, 200 µL from each well was further used for serial dilution in 0.9% saline and spot plating on TSA plates. The TSA plates were then incubated at 37 °C for 24 h for enumeration. The cell counts were reported in CFU/well, and log_10_ CFU reductions were calculated.

### 2.7. Statistical Analysis

The statistical analyses were carried out using GraphPad Prism version 9.5.1. The results were analyzed via a 2-way ANOVA, and *p*-values were corrected for multiple comparisons using Dunnett’s method. The antibiofilm effect of the aPDD condition against food contact surfaces was analyzed via a 2-way ANOVA, and *p*-values were corrected for multiple comparisons using Tukey’s test.

## 3. Results

### 3.1. Screening of Photosensitizer Candidates for Their Antibiofilm Efficacy against S. enterica, MRSA, P. fragi, and B. thermosphacta

For screening, the peak absorption wavelengths (λ_max_) of each PS candidate were analyzed and are listed in Table 1. In cases where multiple peaks were observed, the highest λ_max_ was selected for the screening of the antibiofilm efficacy of the PS candidates against *S. enterica*, MRSA, *P. fragi*, and *B. thermosphacta*. The recovery of biofilms varied for each organism. For example, the highest recoveries from untreated biofilms indicated as “VC dark” (i.e., unirradiated vehicle control) for *S. enterica*, MRSA, *P. fragi*, and *B. thermosphacta* were approximately 4.0, 5.8, 4.1, and 1.3 log_10_ CFU/peg of the MBEC device, respectively (Figure 2, Figure 3, Appendix A), although the initial inoculation density was 10^5^–10^6^ CFU/mL for each organism. This shows that MRSA and *B. thermosphacta* formed the most and the least robust biofilms on polystyrene pegs, respectively, while *S. enterica* and *P. fragi* were intermediate in this regard. All 32 PS candidates were tested on biofilms formed by *S. enterica*, MRSA, *P. fragi*, and *B. thermosphacta* and were screened for their antibiofilm efficacy based on multiple characteristics, such as the formation of a uniform solution in their respective solvents, stability of the solution formed (i.e., no precipitation during storage), and ability to reduce the biofilms by at least 3 log_10_ CFU/peg—or, where the dark VC yielded less than 3 log_10_ CFU/peg, a “max kill” (corresponding to no colonies obtained from the treated biofilms). Note that only the PS candidates meeting the aforementioned criteria are summarized in Table 2, Figure 2, Figure 3, Appendix A; semiquantitative (i.e., turbidimetric) results for the others are summarized in Appendix A.

Methylene Blue (MB), used as the positive control, demonstrated excellent bactericidal and antibiofilm efficacy with values ≤0.001% (*w*/*v*) (i.e., the lowest tested concentration) for both the MBC and MBEC against all four organisms (Table 2), whether treated with 660 nm (deep red) or white LED irradiation. This observation was further confirmed by the quantitative analysis, wherein MB yielded a “max kill” for all four tested organisms. For instance, 0.001% (*w*/*v*) MB resulted in reductions of 4.38, 3.50, 4.05, and 2.11 log_10_ (CFU/peg) in *S. enterica*, MRSA, *P. fragi*, and *B. thermosphacta*, respectively, when treated with a 76.8 J/cm^2^ (8 min) dose of deep red light (Figure 2, Figure 3, Appendix A). Similarly, 0.001% (*w*/*v*) MB resulted in 3.82, 3.11, 3.31, and 1.23 log_10_ CFU/peg reductions (corresponding to max kills) in *S. enterica*, MRSA, *P. fragi*, and *B. thermosphacta*, respectively, when treated with 72 J/cm^2^ (8 min) of white light emitted from the LED (Figure 2, Figure 3, Appendix A).

Dark toxicity (i.e., the efficacy of the PS in the absence of light) was not apparent for the PS candidates based on the semiquantitative turbidimetric analysis (Table 2 and Appendix A), although some PSs showed a low level of dark toxicity against selected microorganisms in the quantitative analysis in this study. For example, 0.1% (*w*/*v*) hemoglobin yielded reductions of 2.05 and 2.69 log_10_ CFU/peg against *S. enterica* and *P. fragi* biofilms when compared with the dark VCs (Figure 2C and Appendix A). Additionally, 0.01% (*w*/*v*) tomato lycopene extract showed dark toxicity against *P. fragi* biofilms, which reduced by 1.14 log_10_ CFU/peg as compared to the dark VCs (Appendix A). Other PS candidates did not show any significant dark toxicity against biofilms formed by *S. enterica*, MRSA, *P. fragi*, and *B. thermosphacta*. Beyond dark toxicity, some of the tested LEDs showed antibiofilm efficacy in the absence of PSs. For example, the treatment of *S. enterica* and *P. fragi* with a 420 nm (violet) LED with a 69.6 J/cm^2^ dose (~8 min) showed no visible growth in the vehicle controls for both the outgrowth (MBC) and recovery plates (MBEC) based on the semiquantitative turbidimetric analysis (Table 2). Similar observations were recorded for the treatment of *S. enterica*, MRSA, *P. fragi*, and *B. thermosphacta* with a 57.6 J/cm^2^ dose with UVA light (365 nm) (Table 2). Based on the quantitative analysis, a 69.6 J/cm^2^ dose of a 420 nm (violet) LED reduced the *S. enterica*, MRSA, *P. fragi*, and *B. thermosphacta* by 3.27, 2.98, 3.56, and 0.65 log_10_ CFU/peg, respectively (Figure 2K, Figure 3K, Appendix A). Similarly, a 365 nm (UVA) LED treatment alone at a 57.6 J/cm^2^ dose resulted in reductions of 4.06, 3.35, 3.43, and 2.10 log_10_ CFU/peg reduction in *S. enterica*, MRSA, *P. fragi*, and *B. thermosphacta* cell counts, respectively (Figure 2L, Figure 3L, Appendix A). Also, some antibiofilm efficacy of the 445 nm indigo LED was also observed in the case of *P. fragi* (Appendix A).

The semiquantitative MBC and MBEC values of PS candidates that showed promising antibiofilm efficacy against the organisms tested are shown in Table 2, while the semiquantitative MBC and MBEC values of other PSs are listed in Appendix A. Several PS candidates showed equivalent bactericidal and antibiofilm efficacies (i.e., growth inhibition at similar concentrations, suggesting that biofilm inhibition occurs via a bactericidal mechanism) when irradiated with their corresponding LED wavelengths against *S. enterica* and MRSA, while the rest yielded visible turbidity for all the tested concentrations (Table 2 and Appendix A). For instance, the same MBC and MBEC values were obtained as 0.01% (MBC) and 0.01% (*w*/*v*) (MBEC) for erythrosin B (527 nm, green), allura red (505 nm, cyan), ponceau SX (505 nm, cyan), and sunset yellow (470 nm, blue) and 0.001% (MBC) and 0.001% (*w*/*v*) (MBEC) for riboflavin (445 nm, indigo), curcumin (420 nm, violet), and β-carotene (420 nm, violet), respectively, when irradiated with an LED emitting light of specific wavelengths for 8 min. Similarly, saffron (470 nm, blue) and riboflavin (445 nm, indigo) yielded the same values for MBC and MBEC in the case of P. fragi when irradiated with LEDs emitting light of the specific wavelengths for 8 min (Table 2 and Appendix A). However, only enocyanin yielded the same MBC and MBEC values against *B. thermosphacta* when irradiated with a 527 nm (green) LED for 8 min (Appendix A). Some PSs showed equivalent bactericidal and antibiofilm efficacies when treated with the white LED. Erythrosin B, when irradiated with the white LED, showed the same MBC and MBEC values in the case of *S. enterica*, MRSA, and *P. fragi*. Also, caramel yielded the same MBC and MBEC values in the case of *S. enterica* and MRSA (Table 2). Curcumin and shikonin yielded the same MBC and MBEC values upon irradiation with the white LED in the case of MRSA and *P. fragi*, respectively (Table 2).

White LED treatments in combination with the appropriate PS candidates resulted in a better antibiofilm effect against both *S. enterica* and MRSA as compared to the PS alone, except for the PS candidates allura red, ponceau sx, sunset yellow, and lycopene in the case of *S. enterica* (Figure 2E,F,H,I) and sunset yellow and lycopene in the case of MRSA (Figure 3H,I), indicating the requirement for the LED rays emitting a wavelength of the maximum absorption peak for these PS candidates. In some cases, white LED light showed lower MBC and MBEC values than tests wherein the PSs were irradiated at their respective peak absorbance wavelengths, indicating better bactericidal and antibiofilm efficacy in a strain-specific manner. However, only caramel yielded superior MBC and MBEC values when irradiated with the white LED against all four tested microorganisms. On the contrary, acid red 27 (*S. enterica*, *P. fragi*) and protoporphyrin (MRSA, *P. fragi*) resulted in lower MBC and MBEC values in the case of the white LED against two of the four microorganisms (Appendix A). Based on the quantitative analysis, 0.01% *w*/*v* of hemin reduced *S. enterica* cell counts by 2.45 log_10_ CFU/peg when irradiated with the white LED and did not show any log_10_ reduction when irradiated with the 630 nm (red) LED. For MRSA, protoporphyrin and hemoglobin irradiated with the white LED yielded a higher (by ~1.5 log_10_ CFU/peg) log_10_ reduction than the 630 nm (red) LED irradiation (Figure 3). For *P. fragi*, carmine, acid red 27, and enocyanin showed higher antibiofilm efficacy when treated with the white LED as compared to the LEDs emitting specific wavelengths, as maximum log_10_ reductions of 1.39, 3.91 (corresponding to a max kill), and 3.73 (corresponding to max kill) log_10_ CFU/peg were obtained with white irradiation as compared to the 0.32, 1.85, and 0.08 log_10_ CFU/peg reductions obtained with specific wavelength LED irradiation. A “hook” (or “filter effect”) was also observed in some cases, where a higher concentration of a PS candidate interfered with the antimicrobial effect of the aPDD treatment and the dark-colored PS solution resulted in a limited transfer of the light rays through the solution. For example, a 0.1% *w*/*v* concentration of allura red and ponceau SX reduced the antibiofilm effect of the aPDD treatment when compared to a lower concentration of 0.01% *w*/*v* (Figure 2E,F).

Certain PS candidates, like titanium dioxide and turmeric extract, did not form a homogenous solution in any of the solvents and precipitated out of the solution; moreover, they did not exhibit promising antibiofilm efficacy against *S. enterica*, MRSA, and *P. fragi* and were consequently excluded from further consideration. Other PSs, like cyanidin-3-glucoside, betanin, black carrot extract, resveratrol, citrus red 2, canthaxanthin, capsanthin, trans-β- apo-8′-carotenal, saffron, lutein, and β-carotene, did not show any antibiofilm efficacy against any of the four tested organisms and were similarly excluded from further work.

Overall, four PS candidates (e.g., riboflavin, curcumin, erythrosin B, and sunset yellow) were identified and selected, based on their antibiofilm efficacy against all four tested organisms (Figure 2, Figure 3, Appendix A and Table 2), for further optimization work. A reduced panel of representative organisms, *S. enterica* (gram-negative bacteria) and MRSA (gram-positive bacteria), were chosen for subsequent investigations based on their robust biofilm formation during the initial screening phase.

### 3.2. Optimization of Photosensitizer Candidates and LED Treatment of Polystyrene Pegs

For further optimization of PS candidates, erythrosin B, sunset yellow, and curcumin were selected, as mentioned in Section 3.1; riboflavin was replaced with the more water-soluble sodium salt of riboflavin-5′-phosphate (R-5-P), which showed better antibiofilm efficacy than riboflavin against MRSA and *S. enterica* (Appendix A) when treated with a 445 nm LED for 5 min (55.5 J/cm^2^). Three of the PS candidates (sunset yellow, curcumin, and riboflavin-5′-phosphate) were tested individually and in pairwise combinations with one another, while erythrosin B was tested separately. Sunset yellow, curcumin, and riboflavin-5′-phosphate were treated with blue (470 nm) LED light at various doses (10.2, 25.5, and 51 J/cm^2^, corresponding to the treatment times of 1, 2.5, and 5 min, respectively). Erythrosin B was treated with green (527 nm) light, applying treatment times of 2, 5, and 10 min corresponding to light doses of 10.2, 25.5, and 51 J/cm^2^, respectively.

Based on the turbidimetric assay, when tested individually, curcumin did not show bactericidal and antibiofilm efficacy against *S. enterica* and MRSA. For instance, curcumin, when irradiated with a 51 J/cm^2^ dose of 470 nm LED light, showed MBC and MBEC values of >0.001% *w*/*v* (i.e., the highest tested concentration) against *S. enterica* and 0.001% and >0.001% *w*/*v*, respectively, against MRSA (Appendix A). Riboflavin-5′-Phosphate showed MBC values of 0.1% *w*/*v* against *S. enterica* and MRSA, indicating promising bactericidal efficacy against both microorganisms (Appendix A). The MBEC values of riboflavin-5′-phosphate were 0.01% and 0.1% *w*/*v* against *S. enterica* and MRSA, respectively, indicating the strain-dependent antibiofilm efficacy of the combination of riboflavin-5′-phosphate and a 470 nm (blue) LED in this study (Appendix A). Similarly, sunset yellow yielded a higher MBEC value of 0.01% *w*/*v* against MRSA as compared to 0.001% *w*/*v* against *S. enterica*, while the bactericidal effect of the aPDD method was the same against both MRSA and *S. enterica* based on the semiquantitative turbidimetric analysis, indicating the strain-specific antibiofilm efficacy of sunset yellow (Appendix A). The combination of riboflavin-5′-phosphate and curcumin showed MBEC values of 0.1% and 0.0001% (*w*/*v*) against *S. enterica* when treated with a 25.5 J/cm^2^ dose of a 470 nm LED as compared to MBC and MBEC values of 0.1% and 0.01% *w*/*v* for riboflavin-5′-phosphate and >0.001% *w*/*v* for curcumin, when used alone followed by irradiation with a higher dose of 51 J/cm^2^ with a 470 nm LED. Additionally, the MBC values were 0.1% and 0.0001% *w*/*v* for riboflavin-5′-phosphate and curcumin when combined and irradiated with a 10.2 J/cm^2^ light dose of 470 nm (blue) light against MRSA as compared to the values of >1% and >0.001% *w*/*v* concentrations for riboflavin-5′-phosphate and curcumin individually when treated with the same dose, indicating a synergistic bactericidal effect for the two PS candidates (Appendix A). Conversely, riboflavin-5′-phosphate showed an antagonistic effect on the antibiofilm and bactericidal effects of sunset yellow based on the semiquantitative analysis, as the MBC and MBEC values for sunset yellow increased from 0.001% to 0.01% *w*/*v* against *S. enterica* when irradiated with the dose of 51 J/cm^2^ with 470 nm (blue) light (Appendix A).

Obtaining the same values for MBC and MBEC indicates that a treatment condition resulted in both a bactericidal and antibiofilm effect of similar potencies against the microorganisms. For *S. enterica*, the PS candidates riboflavin-5′-phosphate and sunset yellow alone showed lower MBC values than the MBEC values, indicating that the LED irradiation with a 51 J/cm^2^ dose with a 470 nm (blue) LED yielded a better bactericidal effect against planktonic cells than its antibiofilm efficacy (Appendix A). However, the various pairwise combinations of the three PS candidates yielded the same values for both the MBC and MBEC for *S. enterica* (Appendix A), indicating the simultaneous bactericidal and antibiofilm effect of the aPDD method with a 51 J/cm^2^ dose irradiation with a 470 nm LED. For MRSA, riboflavin-5′-phosphate and sunset yellow, when used individually, yielded the same values for MBC and MBEC (Appendix A), indicating that the irradiation of these candidates with a 51 J/cm^2^ dose of a 470 nm (blue) LED had simultaneous bactericidal and antibiofilm effects. Additionally, the pairwise combinations of riboflavin-5′-phosphate and sunset yellow and sunset yellow and curcumin yielded equivalent values for both MBC and MBEC for a 51 J/cm^2^ dose treatment with blue light (Appendix A). However, the combination of riboflavin-5′-phosphate and curcumin showed MBEC values of 0.1% and 0.0001% *w*/*v*, respectively, for treatment with a 51 J/cm^2^ dose of blue (470 nm) light irradiation, while the MBC values were observed as 0.1% and 0.0001% *w*/*v* for the PS candidates when irradiated with a lower dose of 10.2 J/cm^2^ with the blue light emitted from the LED (Appendix A), indicating the synergistic bactericidal efficacy of both the photosensitizers.

Based on the quantitative analysis, riboflavin-5′-phosphate, sunset yellow, and curcumin, when tested individually, yielded significant antibiofilm efficacy upon irradiation with the 470 nm (blue) LED with limited to no dark toxicity observed for all the PS candidates. Riboflavin-5′-phosphate, sunset yellow, and curcumin reduced the *S. enterica* biofilm cell counts by 2.65 (“max kill”), 2.29 (“max kill”), and 3.08 log_10_ CFU/peg, respectively, with quantitative MBEC values of 0.1, 0.001, and 0.0001% (*w*/*v*), respectively, when treated with a 51 J/cm^2^ dose (5 min) of a 470 nm (blue) LED (Figure 4A–C). Similarly, the maximum reductions of 5.36, 5.51, and 4.45 log_10_ CFU/peg in MRSA CFU counts were observed with 0.1, 0.01, and 0.001% (*w*/*v*) concentrations of riboflavin-5′-phosphate, sunset yellow, and curcumin, respectively, when irradiated with a 51 J/cm^2^ (5 min) dose of 470 nm LED light (Figure 5A–C). Overall, the 51 J/cm^2^ dose yielded a significantly higher reduction in log_10_ CFU/peg reduction compared to the 10.2 and 25.5 J/cm^2^ doses. Moreover, the 2-way ANOVA indicated a significant interaction between the PS concentration and the irradiation dose (*p* < 0.0001 for all tested PSs for both *S. enterica* and MRSA).

The various pairwise combinations of riboflavin-5′-phosphate, sunset yellow, and curcumin showed significant antibiofilm efficacy (Figure 4 and Figure 5). The pairwise combination of sunset yellow with either riboflavin-5′-phosphate or curcumin did not improve its antibiofilm efficacy and produced a similar reduction in MRSA as observed with the irradiation of sunset yellow alone with a 470 nm (blue) LED (Figure 5, compare panel B to D or E). Similarly, a combination of riboflavin-5′-phosphate and curcumin did not show any promising synergistic antibiofilm efficacy against MRSA and showed a similar log_10_ reduction to that obtained from riboflavin-5′-phosphate alone (Figure 5, compare panel A to F). On the contrary, specific pairwise combinations of the PSs showed antagonistic, neutral, or synergistic effects against *S. enterica*. The combination of sunset yellow with riboflavin-5′-phosphate yielded an antagonistic effect on the antibiofilm efficacy of the sunset yellow against *S. enterica*, as the quantitative MBEC value of sunset yellow increased from 0.001% to 0.01% (*w*/*v*) when combined with riboflavin-5′-phosphate before irradiation with blue (470 nm) light emitted from the LED (Figure 4, compare panel B to D). Conversely, combining riboflavin-5′-phosphate and curcumin improved their antibiofilm efficacy upon irradiation with a 470 nm (blue) LED against *S. enterica* as a “max kill” was observed with a lower dose of 25.5 J/cm^2^ (2.5 min) (Figure 4, compare panels A and C to panel F). We also observed “hook effects,” i.e., reduced antibiofilm efficacy at the highest tested concentrations for the tested PS candidates when treated individually and in the pairwise combinations depending on the organism tested.

Erythrosin B was irradiated with the LED emitting light of a 527 nm (green) wavelength. Based on the semiquantitative analysis, there was minimal to no effect of the PS alone (i.e., in the absence of irradiation) against both MRSA and *S. enterica*. Based on the semiquantitative analysis, the MBC and MBEC values were 0.01% and 0.01% *w*/*v*, respectively, when treated with a dose of 25.5 J/cm^2^ of 527 nm (green) light emitted from the LED against *S. enterica* (Appendix A). The same values for MBC and MBEC indicate the simultaneous bactericidal and antibiofilm efficacy of the combination of erythrosin B and a 527 nm LED treatment against *S. enterica*. Additionally, the quantitative MBEC value of 0.1% *w*/*v* of erythrosin B yielded a 3.56 log_10_ CFU/peg reduction (corresponding to a max kill) against *S. enterica* upon irradiation with a 25.5 J/cm^2^ dose of a 527 nm LED (Appendix A). However, the semiquantitative analysis yielded MBC and MBEC values of 0.001% and 0.1% *w*/*v* and a quantitative MBEC value of 0.01% *w*/*v* in the case of MRSA, yielding a reduction of 5.12 log_10_ CFU/peg upon irradiation with a 10.2 J/cm^2^ dose with a 527 nm LED (Appendix A). These observations indicate that MRSA was more sensitive towards the combination of erythrosin B and a 527 nm LED treatment as compared to *S. enterica*.

### 3.3. LED Treatment of Representative Food Contact Surfaces Challenged with Riboflavin-5′-Phosphate

Selected hard surfaces were inoculated with 10^5^ CFU/mL inoculum, which yielded a biofilm recovery in the range of 3.4 to 7.4 log_10_ CFU/peg for *S. enterica* and from 3.8 to 6.8 log_10_ CFU/peg for MRSA. The recovery was the highest in the case of wood, probably due to its porous nature and the ability of the bacteria to form biofilms in such surfaces [40]. MRSA- and *S. enterica*-inoculated aluminium, stainless steel, galvanized steel, mild steel, copper, and wood were treated with the combination of 0.1% *w*/*v* riboflavin-5′-phosphate and a 55.5 J/cm^2^ dose of indigo (445 nm) LED light (corresponding to the peak absorption wavelength of R-5-P). *S. enterica* experienced its highest reduction of 5.96 log_10_ CFU/well in the case of copper (which corresponds to a “max kill”) and its lowest reduction of 0.35 log_10_ CFU/well in the case of wood, relative to the vehicle control-treated coupons exposed to the dark (“dark VC”) for 10 min (Figure 6A). Similarly, MRSA cell counts were reduced by 6.45 log_10_ CFU/mL (corresponding to a max kill) on copper and by 0.64 log_10_ CFU/mL reduction in the case of wood (Figure 6B) as compared to the dark VC. The combination of riboflavin-5′-phosphate and the 445 nm (indigo) LED treatment achieved a “max kill” for *S. enterica* in the case of all surfaces except for aluminum and wood (Figure 6A). Similarly, the R-5-P+/− indigo light combination yielded a “max kill” for MRSA in the case of all the surfaces except for wood (Figure 6B).

Overall, the biofilm formation of the bacteria varied based on the characteristics of the hard surfaces employed in this study. Furthermore, the antibiofilm efficacy of the riboflavin-5′-phosphate and 445 nm (indigo) LED was influenced by the structural attributes of the surfaces, displaying limited effectiveness against biofilms formed on a porous material (wood) but excellent efficacy against biofilms formed on the other tested surfaces. This study highlighted the promising antibiofilm potential of combining riboflavin-5′-phosphate and 445 nm LED light to combat MRSA and *S. enterica* biofilms formed on various hard surfaces.

## 4. Discussion

### 4.1. Screening Candidates for Antibiofilm Efficacy

In this study, we aligned our approach with the One Health perspective, considering the potential for preventing the spread of zoonotic pathogens to personnel working in food processing facilities as well as consumers, while offering a food contact surface disinfecting strategy that does not promote antimicrobial resistance. Conventional food processing sanitization practices have potential disadvantages; therefore, the further development of aPDD technology for use within the food industry has the potential to improve sanitization practices, effectively manage biofilms, and simultaneously contribute to antimicrobial resistance avoidance.

To achieve this goal, we utilized a high-throughput screen based on the MBEC assay [37], wherein 32 food-safe PS candidates were tested in combination with monochromatic LEDs emitting light at peak spectral emission matching the PS peak absorption wavelength or by using white LED light. The PS candidates were selected based on being food safe and included some natural products (e.g., hemoglobin, hemin, black carrot extract, protoporphyrin IX) as well as several Health Canada-approved food coloring agents [41]. Food-safe colorants were excluded if they were likely to stain food products with an undesirable color (e.g., blue, black). Unlike previous investigations, which often focused on the antimicrobial efficacy of a limited set of (e.g., two or three) PSs against planktonic cells [29,42], the present study is, to our knowledge, the first comprehensive screening of a large panel of food-safe PS candidates for efficacy against both planktonic bacteria and biofilms. Our screening method facilitated testing a range of concentrations of multiple PS candidates simultaneously in 96-well microtiter plates and aided in identifying 4 out of 32 PS candidates for further optimization.

As part of our methodology, we looked at the entire spectrum of visible and UV-A LED light and identified a key trend: antibiofilm efficacy increased as the wavelength decreased; this observation is consistent with other work [43,44]. We tested LEDs emitting light of selected wavelengths in the ultraviolet A and the visible range in combination with the 32 PS candidates (Table 1). We observed that, as the wavelength of light emitted from the LED decreased, there was an increase in the antibiofilm effect against the microorganisms with the LED light alone in the absence of the PS candidates. The most potent antibiofilm effect was observed for UV-A (365 nm) and violet (420 nm) LED treatments, with some efficacy for the indigo (445 nm) treatment (Section 3.1). The antibiofilm effect of LED light alone could be attributed to the excitation of the endogenous pigments with photosensitizing activity, such as porphyrins and flavins [44,45]. Previously, *Salmonella enterica* sv. Typhimurium biofilms were reduced by 4.2 log_10_ CFU/mL after the 455 nm (blue) LED treatment with a dose of 87.3 J/cm^2^ (10 min) on each surface of the SS coupons [46]. Similarly, a 265 nm (UVC) LED treatment with a dose of 1.8 mJ/cm^2^ yielded a reduction of 2 log_10_ CFU/mL in *P. aeruginosa* biofilms on polycarbonate coupons [47]. Nonetheless, the antibiofilm and antimicrobial efficiency was most potent when light was combined with certain PS candidates, and the use of an effective PS has been demonstrated to significantly reduce the intensity and duration of light exposure required for this technology to be deployed in an industrial setting [48,49]. In this study, we observed the same values for MBC and MBEC in the turbidimetric analysis in some cases (Section 3.1), which showed the equivalent bactericidal and antibiofilm efficacy of the aPDD condition, indicating not just bacterial removal from the MBEC pegs during the challenge step but actual eradication.

We selected a panel of microorganisms to test the efficacy of aPDD, including the representative pathogens *S. enterica* (gram-negative bacterium) and MRSA (gram-positive bacterium) [4], as well as the food spoilage bacteria *P. fragi* (gram-negative bacterium) and *B. thermosphacta* (gram-positive bacterium) [9,10,11]. We selected one gram-negative and one gram-positive species for each group (i.e., pathogens and food spoilage bacteria), resulting in a representative and broad panel with which to explore a range of potential aPDD conditions. We selected a drug-resistant strain of *Staphylococcus* (i.e., MRSA) as it is a pathogen of concern within the context of One Health [50]. Furthermore, we wanted to replicate existing medical surface data for photodisinfection, as MRSA can lead to illness in humans upon transfer from these surfaces [51,52]. Additionally, MRSA is a pathogen of concern on food contact surfaces, including in meat packing plants [53]. *Pseudomonas* species are predominantly associated with carcasses and aerobically packaged meat, while *B. thermosphacta* is a common organism associated with slaughterhouses, meat carcasses, and in a variety of packaged meat, including, aerobic-, vacuum-, and modified atmospheric-packaged meat [9,10]. Future work could include evaluating the aPDD efficacy against other relevant microorganisms.

### 4.2. Optimization: Selection of four PS Candidates for Further Exploration

To refine our approach, we concentrated on two microorganisms, *S. enterica* (gram-negative) and MRSA (gram-positive). These were selected based on the robust biofilm recovery obtained during the initial screening and the antibiofilm efficacy produced by the PS candidates under irradiation with their corresponding peak absorption wavelength.

The PS candidates selected for optimization (e.g., erythrosin B, sunset yellow, curcumin, and riboflavin-5′-phosphate) underwent further scrutiny to determine the optimal usage concentration and irradiation durations (i.e., equivalent to irradiation dose) and to test for any synergy of pairwise combinations. For the optimization of riboflavin, a water-soluble phosphate sodium salt, riboflavin-5’-phosphate, was used instead of riboflavin due to its higher water solubility, common use as a nutritional supplement, and better antibiofilm efficacy against MRSA and *S. enterica* (Appendix A). The four PS candidates were selected based on the following parameters: the formation of a homogenous solution in their respective solvents, the stability of the solution formed, and their ability to reduce the biofilms of all four tested organisms by at least 3 log_10_ CFU/peg or a “max kill”. As these PSs did not produce dark toxicity in this study, the photochemical mechanism for antibiofilm efficacy could be the same for the four PS candidates, i.e., photodynamic inactivation, which is based on the generation of ROS that lead to cytotoxic responses in bacteria, eventually causing cell death [30].

The antibiofilm efficacy of the PS + light combinations could be attributed to the concentration of the PS used, the light parameters, and the strain used (Section 3.2). A species-specific dose response has been previously shown against planktonic cells, where the dose is the amount of light energy delivered over a certain area of treatment [54,55]. Similarly, the effect of varying the concentration of the PS has been reported in previous studies [56,57]. We also observed a “hook” (or “filter”) effect in this study wherein a higher concentration of some PSs (e.g., riboflavin-5′-phosphate, sunset yellow) yielded a smaller reduction in *S. enterica* or MRSA cell counts (Figure 4A,D,F and Figure 5B–F). The high concentration of the colored PS candidates limited the penetration of the light radiation through the solution, limiting its antimicrobial efficacy. We also observed the filter effect in Figure 2D–F and Figure 3B. This effect was more pronounced in the case of the pairwise combination of riboflavin-5′-phosphate and sunset yellow than the combination of riboflavin-5′-phosphate and curcumin (Figure 4D,F). This can most likely be attributed to the intense light absorption afforded by the higher concentrations of these PSs, which limits the penetration depth of the light into the solution [58]. Therefore, factors like the strain tested, concentration of the PS used, treatment dose, treatment time, and wavelength of the light used influence the antimicrobial effect. Thus, light and PS dosimetry parameters used for aPDD must be carefully considered and optimized when scaling up this technology for application in the food industry.

The antibiofilm photodynamic efficacy of subsets of the four selected PS candidates is described for the first time in this study. To our knowledge, there have been no published studies evaluating the photodynamic efficacy of sunset yellow. Previous studies of erythrosin B focused on evaluating its antimicrobial effect against planktonic cells in combination with LED light, while other studies include the antibiofilm effect of erythrosin B in combination with light emitted from laser diodes [59] and tungsten filament lamps [60]. For instance, 1 mg/mL of erythrosine irradiated with a tungsten filament lamp emitting white light reduced *Streptococcus mutans* biofilms formed on a constant-depth film fermenter by 2.2 log_10_ CFU [60]. The present study is novel as it focuses on the antibiofilm effect of erythrosin B combined with a LED light source. There are reports citing the antibiofilm effect of curcumin with LED light for medical applications, like the reduction in MRSA biofilms formed in bone cavities when treated with a 20.1 J/cm^2^ dose of a blue LED (450 nm) in combination with curcumin [61,62]. The combination of curcumin and a UV lamp has been evaluated against planktonic cells relevant for food safety [57]. Additionally, there have been studies that focused on the antimicrobial efficacy of curcumin in combination with LED light [63,64,65,66]. Previous studies have evaluated the antibacterial effect of the combination of curcumin with lauric arginate ethyl ester micelles against *E. coli* and *Listeria innocua* when irradiated with a UV-A (365 nm) lamp [67,68]. However, we evaluated the antibiofilm effect of the aPDD condition comprising the pairwise combinations of curcumin and PSs, like riboflavin-5′-phosphate or sunset yellow. Pairwise combinations of two photosensitizers for an antibiofilm effect represents a novel approach. Previous studies have shown a limited antibiofilm effect of riboflavin-5′-phosphate upon irradiation with LED light on surfaces other than food contact surfaces. Leelanarathiwat et al. [69] reported that the combination of 0.18 mg/mL of riboflavin-5′-phosphate with a 37–40 J/cm^2^ dose of blue LED light yielded a 1.23 log_10_ CFU/mL reduction in *S. aureus* biofilms as compared to the >3 log_10_ CFU/mL reduction in MRSA and *S. enterica* cell counts observed in this study, which might have been achieved as a result of using a higher dose (55.5 J/cm^2^) of indigo (445 nm) light.

The various pairwise combinations of riboflavin-5′-phosphate, sunset yellow, and curcumin showed either synergistic, additive, or antagonistic effects against *S. enterica* biofilms in this study. For instance, riboflavin-5′-phosphate + sunset yellow showed antagonistic effect on the antibiofilm effect of sunset yellow, while riboflavin-5′-phosphate + curcumin showed synergistic antibiofilm efficacy against *S. enterica* (Section 3.2). There have been limited studies focusing on combinations of PSs. In previous reports, combinations of PSs have shown improved antimicrobial efficacy but require increased PS concentration as compared to the concentrations required when used alone [70]. Interestingly, the combinations of chlorin compounds with exogenous porphyrin compounds showed an antagonistic effect on the antimicrobial efficacy of the individual PS [70].

### 4.3. LED Treatment of Food Contact Surfaces Challenged with Riboflavin-5′-Phosphate

The present study has demonstrated the aPDD efficacy of a range of PSs, but for the results to be applicable in a food processing setting, the efficacy must be trialed on appropriate surfaces. Therefore, as a proof of concept for the present study, we tested the ability of the combination of 0.1% (*w*/*v*) riboflavin-5′-phosphate and a 445 nm (indigo) LED treatment at 55.5 J/cm^2^ to eliminate *S. enterica* and MRSA biofilms on a range of contact surfaces. The surfaces were selected due to their common use in food processing facilities or in other food-related settings and included aluminum, stainless steel, galvanized steel, mild steel, copper, and wood. Impressively, R-5-P + indigo light yielded a “max kill” on most of the tested surfaces except for wood. The potency observed here (approximately a 5 log_10_ CFU reduction) is comparable to the antibiofilm efficacy of existing chemical sanitizers used in the food industry, like quaternary ammonium compounds and triclosan [71], but without the disadvantages of being corrosive and contributing to microbes developing resistance [19].

The minimal efficacy against biofilms formed in the case of wood could be due to the shadowing effect caused by material porosity and rugosity, i.e., the 445 nm LED treatment could not reach the bacterial cells present in the crevices of the irregular surface of the wood. This finding demonstrates the limitation of aPDD to eliminate biofilms on rough surfaces that might obscure light illumination. To address this challenge, the aPDD approach might have to be further optimized for wood surfaces (e.g., using a higher light energy or incorporating multiple angles of irradiation). Future efforts should involve evaluating aPDD efficacy against other suitable foodborne microorganisms and using additional food contact surfaces (e.g., plastics, such as High-Density Polyethylene (HDPE), polyurethane, vinyl, Polytetrafluoroethylene (PTFE), or Polypropylene (PP)), along with rigorous testing within actual food processing facilities.

## 5. Conclusions

This work demonstrated the in vitro antimicrobial efficacy of various food-safe PS candidates against a representative panel of foodborne pathogens. We optimized the process for a more focused set of PS candidates, allowing us to delve deeper into the efficacy of promising PS candidates from the screening phase and offer enhanced insights into their potential practical applications. Riboflavin-5′-phosphate in combination with 445 nm (indigo) LED light showed a promising antimicrobial effect against the tested microorganisms on food contact surfaces and has the potential to be used as a no-rinse photoantimicrobial for the food industry. The antimicrobial kinetics of aPDD depend on the light and PS dosimetry parameters, as well as the bacterial species used. Although our aPDD protocol against biofilms grown on plastic or metal surfaces was highly effective, it did not effectively inactivate biofilms formed on wood, likely due to the porosity of the material, indicating that further optimization is warranted for such surfaces. Overall, aPDD technology represents a promising strategy to extend the shelf life of food and reduce risks of foodborne pathogen exposure to consumers.

## Figures and Tables

**Figure 1 foods-13-00453-f001:**
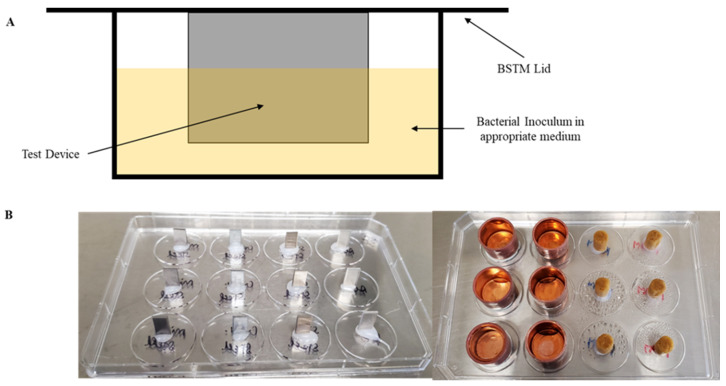
Schematic of a cross-section of an individual well of the BSTM™ device (**A**). Representative images of the food contact surfaces, like aluminum, stainless steel, galvanized steel, mild steel, copper, and wood, affixed onto 12-well lids (**B**).

**Figure 2 foods-13-00453-f002:**
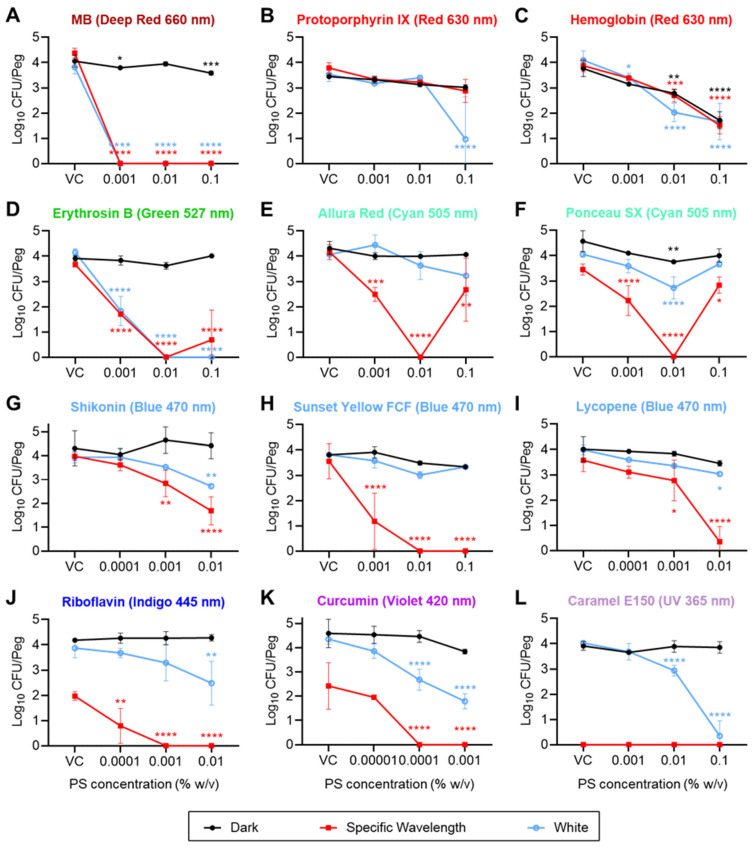
Recovery of viable *S. enterica* from MBEC pegs (log_10_ CFU/peg). Biofilms were challenged with methylene blue (**A**), protoporphyrin IX (**B**), hemoglobin (**C**), erythrosin B (**D**), allura red (**E**), ponceau SX (**F**), shikonin (**G**), sunset yellow FCF (**H**), tomato lycopene extract (**I**), riboflavin (**J**), curcumin (**K**), or caramel color E150 (**L**), at the indicated concentrations (x-axis; VCs = vehicle controls), and subjected to the indicated irradiation conditions: dark (represented by black lines) vs. specific wavelength (indicated above each graph, represented by red lines) vs. white (represented by blue lines). Symbols and error bars represent the mean ± standard deviation for 3 replicate pegs. Statistical significance, as indicated for certain data points, was evaluated via a 2-way ANOVA, and *p*-values (relative to the corresponding “VC” data point for each irradiation condition) were corrected for multiple comparisons using Dunnett’s method. *, *p* < 0.05; **, *p* < 0.01; ***, *p* < 0.001; ****, *p* < 0.0001.

**Figure 3 foods-13-00453-f003:**
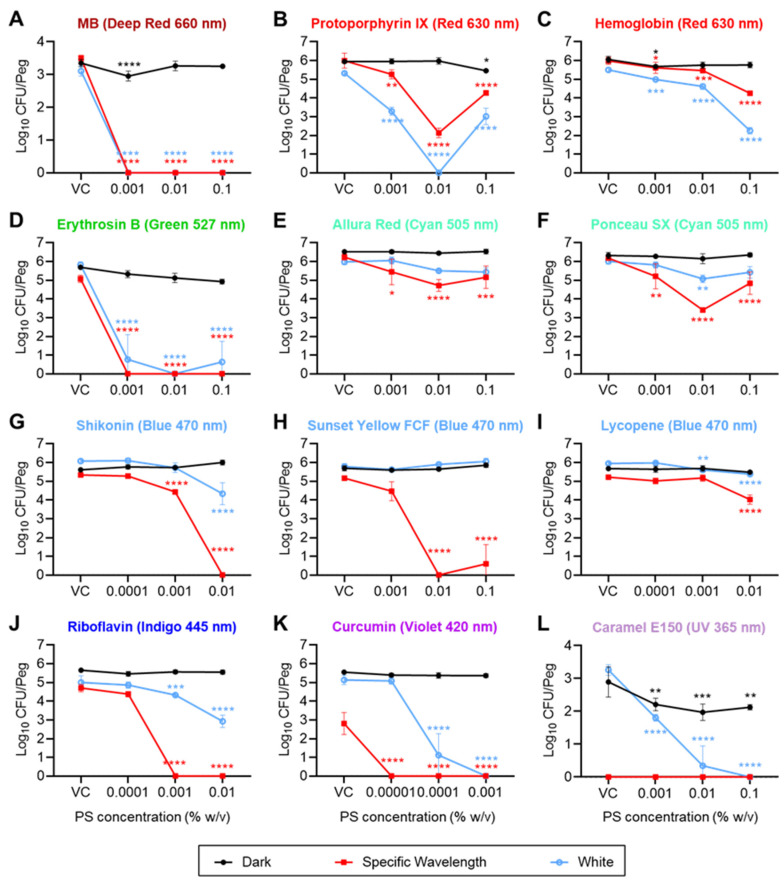
Recovery of viable MRSA from MBEC pegs (log_10_ CFU/peg). Biofilms were challenged with methylene blue (**A**), protoporphyrin IX (**B**), hemoglobin (**C**), erythrosin B (**D**), allura red (**E**), ponceau SX (**F**), shikonin (**G**), sunset yellow FCF (**H**), tomato lycopene extract (**I**), riboflavin (**J**), curcumin (**K**), or caramel color E150 (**L**), at the indicated concentrations (x-axis; VCs = vehicle controls), and subjected to the indicated irradiation conditions: dark (represented by black lines) vs. specific wavelength (indicated above each graph, represented by red lines) vs. white (represented by blue lines). Symbols and error bars represent the mean ± standard deviation for 3 replicate pegs. Statistical significance, as indicated for certain data points, was evaluated via a 2-way ANOVA, and *p*-values (relative to the corresponding “VC” data point for each irradiation condition) were corrected for multiple comparisons using Dunnett’s method. *, *p* < 0.05; **, *p* < 0.01; ***, *p* < 0.001; ****, *p* < 0.0001.

**Figure 4 foods-13-00453-f004:**
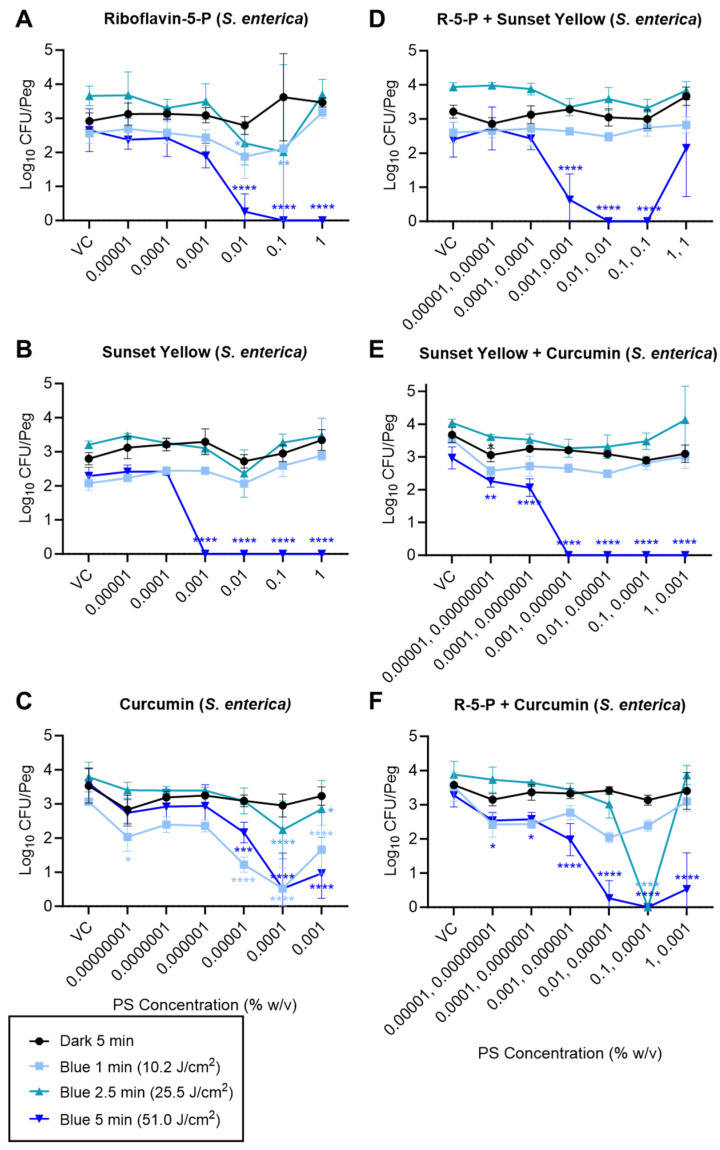
Recovery of viable *S. enterica* from (470 nm) blue-light-irradiated MBEC pegs (log_10_ CFU/peg). Biofilms were challenged with riboflavin-5′-phosphate (**A**), sunset yellow (**B**), curcumin (**C**), riboflavin-5′-phosphate + sunset yellow (**D**), sunset yellow + curcumin (**E**), or riboflavin-5′-phosphate + curcumin (**F**), at the indicated concentrations (x-axis; VCs = vehicle controls), for the indicated irradiation times. Symbols and error bars represent the mean ± standard deviation for 4 replicate pegs. Statistical significance, as indicated for certain data points, was evaluated via a 2-way ANOVA, and *p*-values (relative to the corresponding “VC” data point for each irradiation condition) were corrected for multiple comparisons using Dunnett’s method. *, *p* < 0.05; **, *p* < 0.01; ***, *p* < 0.001; ****, *p* < 0.0001.

**Figure 5 foods-13-00453-f005:**
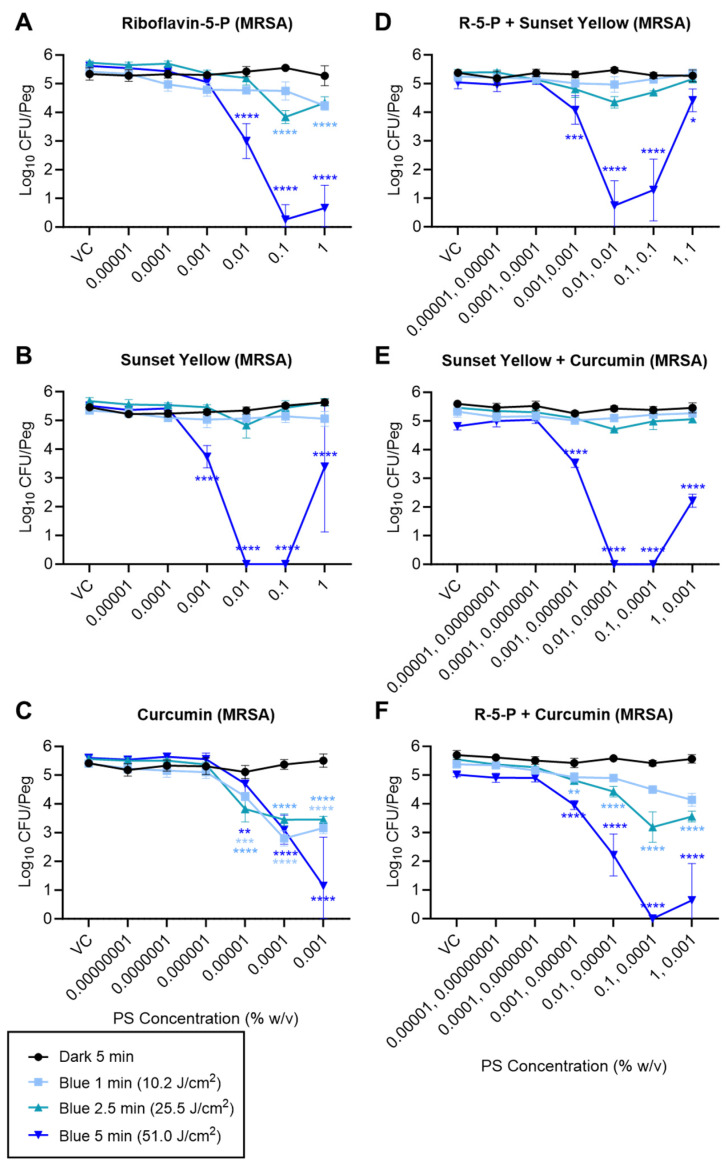
Recovery of viable MRSA from blue-light-irradiated MBEC pegs (log_10_ CFU/peg). Biofilms were challenged with riboflavin-5′-phosphate (**A**), sunset yellow (**B**), curcumin (**C**), riboflavin-5′-phosphate + sunset yellow (**D**), sunset yellow + curcumin (**E**), or riboflavin-5′-phosphate + curcumin (**F**), at the indicated concentrations (x-axis; VCs = vehicle controls), for the indicated irradiation times. Symbols and error bars represent the mean ± standard deviation for 4 replicate pegs. Statistical significance, as indicated for certain data points, was evaluated via a 2-way ANOVA, and *p*-values (relative to the corresponding “VC” data point for each irradiation condition) were corrected for multiple comparisons using Dunnett’s method. *, *p* < 0.05; **, *p* < 0.01; ***, *p* < 0.001; ****, *p* < 0.0001.

**Figure 6 foods-13-00453-f006:**
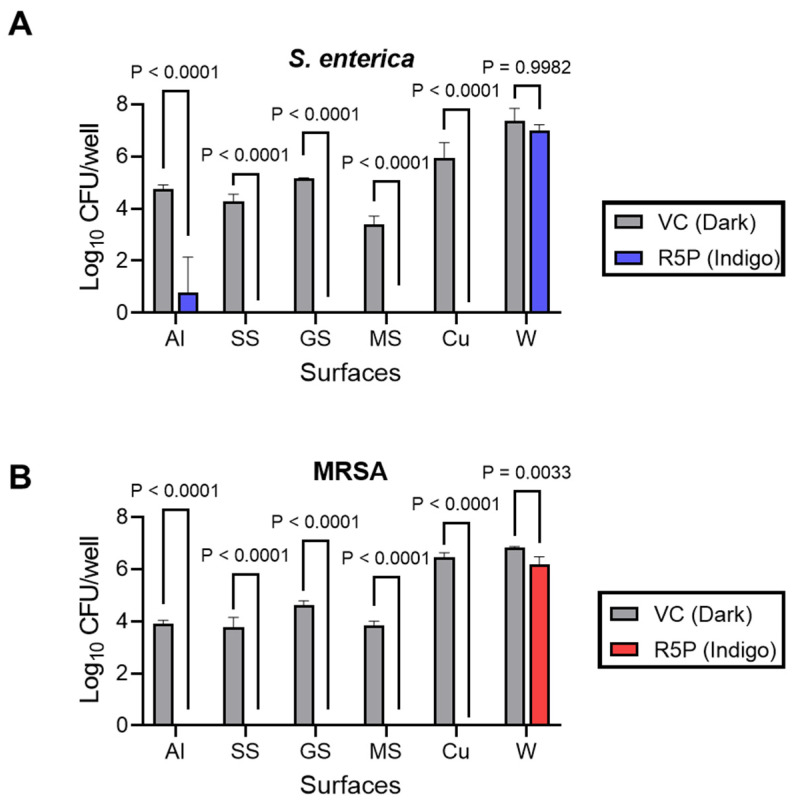
Recovery of viable *S. enterica* (**A**) or MRSA (**B**) from the various tested surfaces, including, aluminum (Al), stainless steel (SS), galvanized steel (GS), mild steel (MS), copper (Cu), and wood (W). The indicated surface was either treated with water (vehicle control, “dark VC”) and incubated in the dark or treated with Riboflavin-5′-phosphate (“R-5-P Indigo”) before irradiation with a 445 nm LED at 55.5 J/cm^2^. Each bar represents the mean ± standard deviation for 3 replicates. Statistical significance was evaluated via a 2-way ANOVA, and *p* values were corrected for multiple comparisons using Tukey’s test.

**Table 1 foods-13-00453-t001:** List of photosensitizers screened for their antimicrobial efficacy, their observed peak absorption wavelength (λ_max_), and the irradiation conditions (wavelength, irradiance, and dose) used in this screening.

Photosensitizer Candidate	Manufacturer, Manufacturer Address	Final Tested Concentrations (% *w*/*v*) ^1^	λ_max_ (nm) ^2^	LED and λ ^3^	Irradiance (mW/cm^2^)	Dose (J/cm^2^)
Methylene Blue (Positive Control)	Ondine Biomedical Inc., Vancouver, BC, Canada	0.1, 0.01, 0.001	**668**	Deep red (660 nm)	160	76.8
Protoporphyrin IX	Sigma Aldrich, St. Louis, MO, USA	0.1, 0.01, 0.001	**643**	Red (630 nm)	125	60.0
Hemin	Sigma Aldrich, St. Louis, MO, USA	0.01, 0.001, 0.0001	**642**			
Hemoglobin	Sigma Aldrich, St. Louis, MO, USA	0.1, 0.01, 0.001	**631**			
Titanium Dioxide	Thermo Scientific, Mississauga, ON, Canada	0.1, 0.01, 0.001	NT ^4^	Amber (590 nm)	130	62.4
Orchil (Orcein)	Thermo Scientific, Mississauga, ON, Canada	0.01, 0.001, 0.0001	**575**			
Cochineal (Carmine)	Millipore sigma, St. Louis, MO, USA	0.01, 0.001, 0.0001	**556**			
Anthocyanin (Cyanidin-3-glucoside)	APExBIO, Houston, TX, USA	0.01, 0.001, 0.0001	**518**	Green (527 nm)	85	40.8
Betanin (Beet Red)	TCI America, Portland, OR, USA	0.1, 0.01, 0.001	**537**			
Black Carrot Extract	Rooted (purchased through ETSY), Coquitlam, BC, Canada	0.1, 0.01, 0.001	**532**			
Erythrosin B	Thermo Scientific, Mississauga, ON, Canada	0.1, 0.01, 0.001	**529**			
Acid Red 27 (Amaranth)	TCI America, Portland, OR, USA	0.1, 0.01, 0.001	**519**			
Enocyanin (an Anthocyanin)	MedchemExpress, Monmouth junction, NJ, USA	0.1, 0.01, 0.001	**520**			
Resveratrol (Grape Skin Extract)	EMD Millipore, Billerica, MA, USA	0.01, 0.001, 0.0001	**315**			
Citrus Red No. 2	Sigma Aldrich, St. Louis, MO, USA	0.01, 0.001, 0.0001	510	Cyan (505 nm)	110	52.8
Allura Red	TCI America, Portland, OR, USA	0.1, 0.01, 0.001	**500**			
Ponceau SX	Sigma Aldrich, St. Louis, MO, USA	0.1, 0.01, 0.001	**500**			
Canthaxanthin	LKT laboratories Inc., St. Paul, MN, USA	0.001, 0.0001, 0.00001	**462**	Blue (470 nm)	170	81.6
Shikonin (Alkanet)	Enzo, Farmingdale, NY, USA	0.01, 0.001, 0.0001	**483**			
Sunset Yellow FCF	TCI America, Portland, OR, USA	0.1, 0.01, 0.001	**481**			
Tomato Lycopene Extract	USP, Rockville, MD, USA	0.01, 0.001, 0.0001	**477**			
Paprika (Capsanthin)	LKT laboratories Inc., St. Paul, MN, USA	0.01, 0.001, 0.0001	**462**			
Trans-β-apo-8′-carotenal	Sigma Aldrich, Burlington, MA, USA	0.001, 0.0001, 0.00001	**443**			
Saffron	Sigma Aldrich, St. Louis, MO, USA	0.1, 0.01, 0.001	**469**			
Xanthophyll (Lutein)	Cayman Chemical Co., Ann Arbor, MI, USA	0.01, 0.001, 0.0001	**462**	Indigo (445 nm)	185	88.8
Ethyl β-apo-8′-carotenoate	US Biological Life Sciences, Swampscott, MA, USA	0.001, 0.0001, 0.00001	**428–502**			
Riboflavin	Sigma Aldrich, St. Louis, MO, USA	0.01, 0.001, 0.0001	**443**			
Curcumin	Cayman Chemical Co., Ann Arbor, MI, USA	0.001, 0.0001, 0.00001	**428**	Violet (420 nm)	145	69.6
Beta-Carotene	TCI America, Portland, OR, USA	0.001, 0.0001, 0.00001	**425 **			
Tartrazine (Acid Yellow 23)	TCI America, Portland, OR, USA	0.1, 0.01, 0.001	**425**			
Norbixin (Annatto)	Toronto Research Chemicals inc., Toronto, ON, Canada	0.01, 0.001, 0.0001	**423**			
Turmeric Extract	McCormik Club House, London, ON, Canada	0.001, 0.0001, 0.00001	**424**			
Caramel E150	Hunan Jq material Tech Co., ltd., Chansha, Hunan, China	0.1, 0.01, 0.001	**294**	UV (365 nm)	120	57.6

^1^ All potential photosensitizers were prepared at a “master stock” concentration in either water (blue font), DMSO (red font), or propylene glycol (green font). Each of these master stocks were diluted in water to produce “working stocks”, which were further diluted in water (1:10 and 1:100). These final challenge concentrations are reported here. ^2^ The absorbance was measured over the wavelength range of 200–800 nm using a UV-Vis spectrophotometer. Where more than one wavelength could potentially be tested, one was chosen (indicated in bold font)—typically, the highest wavelength yielding an obvious peak, or the wavelength yielding an obviously predominant peak. In the case of citrus red no. 2, the wavelength of 510 nm was chosen based on a published paper [36]. ^3^ In addition to irradiation with the specified light source, replicate plates were incubated in the dark (control) or irradiated with white LED light (LUM296LAWHT; 150 mW/cm^2^; 72.0 J/cm^2^). ^4^ NT = not tested (formed insoluble crystals) and used instead as a suspension in water (10 mg/mL).

**Table 2 foods-13-00453-t002:** The minimum bactericidal concentration (MBC) and minimum biofilm eradication concentration (MBEC) values (in % *w*/*v*) of the selected photosensitizers, based on the turbidimetric assay for *S. enterica*, MRSA, *P. fragi*, and *B. thermosphacta*.

Photosensitizer ^a^	LED	Turbidimetric Assay							
		*S. enterica*		MRSA		*P. fragi*		*B. thermosphacta*	
		MBC	MBEC	MBC	MBEC	MBC	MBEC	MBC	MBEC
Methylene Blue	Dark	0.1	>0.1	0.01	>0.1	0.01	0.1	0.01	>0.1
	Deep red	≤0.001	≤0.001	≤0.001	≤0.001	≤0.001	≤0.001	≤0.001	≤0.001
	White	≤0.001	≤0.001	≤0.001	≤0.001	≤0.001	≤0.001	≤0.001	≤0.001
Hemoglobin	Dark	>0.1	>0.1	>0.1	>0.1	>0.1	>0.1	N/A	N/A
	Red	>0.1	>0.1	>0.1	>0.1	>0.1	>0.1	N/A	N/A
	White	>0.1	>0.1	>0.1	>0.1	>0.1	>0.1	N/A	N/A
Erythrosin B	Dark	>0.1	>0.1	>0.1	>0.1	>0.1	>0.1	>0.1	>0.1
	Green	0.01	0.01	≤0.001	≤0.001	≤0.001	≤0.001	≤0.001	≤0.001
	White	0.01	0.01	≤0.001	≤0.001	0.001	≤0.001	≤0.001	≤0.001
Allura Red	Dark	>0.1	>0.1	>0.1	>0.1	>0.1	>0.1	>0.1	>0.1
	Cyan	0.01	0.01	>0.1	>0.1	>0.1	>0.1	0.01	N/A
	White	>0.1	>0.1	>0.1	>0.1	>0.1	>0.1	0.01	0.01
Ponceau SX	Dark	>0.1	>0.1	>0.1	>0.1	>0.1	>0.1	>0.1	>0.1
	Cyan	0.01	0.01	>0.1	>0.1	>0.1	>0.1	0.01	N/A
	White	>0.1	>0.1	>0.1	>0.1	>0.1	>0.1	0.001	0.001
Shikonin	Dark	>0.01	>0.01	>0.01	>0.01	>0.01	>0.01	>0.01	>0.01
	Blue	>0.01	>0.01	0.01	0.01	0.01	0.001	0.001	N/A
	White	>0.01	>0.01	>0.01	>0.01	0.01	0.01	0.01	≤0.0001
Sunset Yellow	Dark	>0.1	>0.1	>0.1	>0.1	>0.1	>0.1	>0.1	>0.1
	Blue	0.01	0.01	0.01	0.01	0.01	≤0.001	≤0.001	≤0.001
	White	>0.1	>0.1	>0.1	>0.1	>0.1	>0.1	0.01	N/A
Lycopene	Dark	>0.01	>0.01	>0.01	>0.01	>0.01	>0.01	>0.01	>0.01
	Blue	>0.01	>0.01	>0.01	>0.01	0.01	0.001	0.001	N/A
	White	>0.01	>0.01	>0.01	>0.01	>0.01	0.01	0.01	N/A
Riboflavin	Dark	>0.01	>0.01	>0.01	>0.01	>0.01	>0.01	>0.01	0.01
	Indigo	0.001	0.001	0.01	0.001	0.001	0.001	0.001	N/A
	White	>0.01	>0.01	>0.01	>0.01	0.01	>0.01	>0.01	0.01
Curcumin	Dark	>0.001	>0.001	>0.001	>0.001	>0.001	>0.001	>0.001	0.001
	Violet	0.0001	≤0.0001	0.0001	0.00001	0.0001	N/A	N/A	0.0001
	White	>0.001	>0.001	0.001	0.001	0.001	0.0001	0.0001	N/A
Caramel	Dark	>0.1	>0.1	>0.1	>0.1	>0.1	>0.1	>0.1	>0.1
	UV	N/A	N/A	N/A	N/A	N/A	N/A	N/A	N/A
	White	0.1	0.1	0.1	0.1	0.1	0.01	0.1	0.01

^a^ The photosensitizers were prepared at a “master stock” concentration in either water (blue font) or DMSO (red font). N/A = Not Applicable (no growth obtained in at least 2 of 3 VC wells).

## Data Availability

All data generated or analyzed during this study are included in this published article and its Appendix A.

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
