# Peer review of "Photodynamic Inactivation of Foodborne Bacteria: Screening of 32 Potential Photosensitizers"

_foods, 2024, doi:10.3390/foods13030453_

Round 1

Reviewer 1 Report

Comments and Suggestions for Authors

The study on photodynamic inactivation of foodborne bacteria with various potential photosensitizers is thorough, but some aspects could be enhanced or areas where its limitations are apparent:

1. The writing in the introduction section could be further improved.

2. Methodology in "2.4. Screening of photosensitizer candidates against biofilms and planktonic cells": For a more advanced research approach, it is recommended to refer to the publication "J. Agric. Food Chem. 2022, 70, 7547−7565 (https://doi.org/10.1021/acs.jafc.2c01667)".

3. In Figure 5, additional explanations are needed regarding the "hook" (or "filter") effect observed in Figure 5.

4. The study focuses on 32 food-safe pigments, including some approved by Health Canada. However, limiting the range of photosensitizers to only food pigments may restrict the potential of the study. Exploring a broader range of photosensitizers could yield more effective or efficient choices.

5. Although the study suggests that pathogens do not develop resistance to photodynamic inactivation, this conclusion is based on short-term observations. Long-term studies are necessary to confirm that repeated use of photodynamic inactivation does not lead to resistance.

6. Although the study employs food colorants approved by Health Canada, the safety implications of using these substances as photosensitizers under light exposure, particularly the safety evaluation of the photodegradation products of these colorants, need to be thoroughly assessed. This is essential for regulatory approval and consumer acceptance.

Comments on the Quality of English Language

The writing in the introduction section could be further improved.

Reviewer 2 Report

Comments and Suggestions for Authors

The topic is within the scope of the Journal, which may interest its readers. The experiments are well-designed and the proposed tests were clearly described. However, the section describing the results could benefit from being shorter. The amount of written content is significantly higher compared to the discussion section. It may strengthen the study's message to remove P. fragi and B. thermosphacta and focus solely on Salmonella and MRSA.

Given the amount of text, the paragraphs summarizing the partial outcomes of the study, such as Lines 642-9, should be avoided and meshed into the discussion part of the manuscript.

Line 796-800. First, you state curcumin is explored for medical applications when it comes to employing LED light and observing antibiofilm effects, and then you cite papers such as 63 revealing its antibiofilm effect from a food safety point of view. However, recent studies have shown that LED light, specifically blue light, can effectively inactivate biofilms of foodborne pathogens like L. monocytogenes. Please elaborate carefully, as there are lots of examples in the literature of using curcumin as a food-grade photosensitizer for decontaminating both food matrices and, more recently right now, food contact surfaces. This also concerns section 4.3, Line 820. The study findings need to be compared with literature on different antimicrobial treatment efficacy depending on the hard surface on which bacteria reside. The paragraph (Line 810-19) regarding the PS combinations is not sufficiently discussed. It is crucial to conduct a thorough literature search for studies that specifically investigate combinations and give specific PS examples, their concentrations, and microbes.

Figure 6. Could you kindly provide the list of acronyms used in the study along with their respective descriptions for the surfaces?

Figure S5 is unclear and probably unnecessary. Nothing is mentioned about it in the manuscript. Tables S1 through S6 as well. Tables S1 through S6 are frequently cited but provide little information, making it difficult for the reader to keep up with the text.

I don't understand how the effect is observed based on the data in Tables S5 and S6 for synergistic/antagonist/neutral.

Reviewer 3 Report

Comments and Suggestions for Authors

This paper is generally well-written and can be a useful reference for various industries.

Some minor revisions are needed within the main body of the text and with an enhanced discussion.

1.     Both summarization and enhancement are needed in the introduction. There is redundant content between the discussion section (lines 687-712) and the introduction. It would be beneficial to transfer the objectives mentioned in the discussion section to the introduction.

2.     There is no description of the fundamental mechanisms behind the roles of light sources, photosensitizers, and the interactions between surface materials or biomolecules. Please consider adding explanations of type I and type II reactions in photodynamic reactions to either the introduction or discussion section.

3.     Line 326-327: Is there any relevance between the surface materials mentioned here and food? Please provide examples explaining why each material was used. Did the authors directly inoculate bacteria on the surface of food and apply photosensitizer treatment along with the light sources? Using the term "food" might typically imply sterilization of the food itself. If authors are referring to packaging materials for food, it would be clearer to use more precise terminology or remove the word "food."

4.     Fig. 2: Why do you think the results increased at 0.1 concentration in Figures D, E, and F? The reasons for this should be clearly documented in the discussion.

5.     Fig. 2: Figures E, F, H, and I do not show significant differences between the white light source and dark. Literature review or discussion regarding this phenomenon is needed.

6.     Have you also tested beneficial bacteria, such as probiotics, with the light sources used in this study? The response of beneficial bacteria is an important issue to consider.

7.     Line 815-819: You explained the antagonistic effect, but what is the underlying mechanism of this phenomenon?

8.     R5P is a substance commonly used in food and cosmetics. What are the potential side effects on the skin or food when exposed to increased sunlight during the summer?

9.     Please add a flowchart of the experimental procedure as a figure.

Reviewer 4 Report

Comments and Suggestions for Authors

Dear Authors,

The work entitled “Photodynamic inactivation of foodborne bacteria: screening of 32 potential photosensitizers” (foods-2821397) is very interesting and necessary in the era of the increasingly difficult elimination of microorganisms resistant to various pesticides.

The work is written clearly. The subsequent planned and implemented steps are fully justified.

Of the 32 photosensitive substances tested, four compounds were selected as candidates for photosensitizers. The selection was done based on their antibiofilm effectiveness against all four tested organisms.

Only Salmonella enterica (Gram-negative bacteria) and MRSA (Gram-positive bacteria) were selected for subsequent tests, which is also justified because they formed the strongest biofilm during the screening phase of the experiments. Researchers also observed both antagonism and synergism of the effects of substances acting on bacteria in combination, which makes the results even more interesting and valuable.

Photodynamic antimicrobial treatment/disinfection is a technology that represents a promising strategy to eliminate bacterial biofilms and in consequence, allows for extended shelf-life and reduces risks of foodborne pathogen exposure to consumers.

In conclusion, the results obtained in the current study are original, interesting, and noteworthy. The Discussion section is very extensive and the results of the work are well-discussed in light of the results of other authors.

-          Line 39 – change C. perfringens to Clostridium perfringens – first use, whole Latin name

-          Line 130 – EP, JP, UPS – all the abbreviations should be explained when introduced; please, check the whole manuscript - e.g. line 842 (HDEP, PU, PTFE, PP)

-          Line 135 – describe briefly the conditions of the sonication process

-          Line 456 – Table 2 – should be Brochothrix thermosphacta instead of Brochothrix Thermosphacta

-          Lines 641, 675 – S. enterica in Italics – check the whole manuscript

-          Please review the entire manuscript and consistently use bacterial species names as abbreviations when the Latin name of a species has been used for the first time.

Round 2

Reviewer 1 Report

Comments and Suggestions for Authors

no comments

Reviewer 2 Report

Comments and Suggestions for Authors

No further comments